# Interfacial interaction and intense interfacial ultraviolet light emission at an incoherent interface

Xuexi Yan[1,6], Yixiao Jiang [1,6], Qianqian Jin[2], Tingting Yao[1], Weizhen Wang[1], Ang Tao[1], Chunyang Gao[1], Xiang Li[1], Chunlin Chen [1,3] ✉, Hengqiang Ye[3] & Xiu-Liang Ma[1,4,5] ✉

Incoherent interfaces with large mismatches usually exhibit very weak interfacial interactions so that they rarely generate intriguing interfacial properties. Here we demonstrate unexpected strong interfacial interactions at the incoherent AlN/Al$_2$O$_3$ (0001) interface with a large mismatch by combining transmission electron microscopy, first-principles calculations, and cathodoluminescence spectroscopy. It is revealed that strong interfacial interactions have significantly tailored the interfacial atomic structure and electronic properties. Misfit dislocation networks and stacking faults are formed at this interface, which is rarely observed at other incoherent interfaces. The band gap of the interface reduces significantly to ~3.9 eV due to the competition between the elongated Al-N and Al-O bonds across the interface. Thus this incoherent interface can generate a very strong interfacial ultraviolet light emission. Our findings suggest that incoherent interfaces can exhibit strong interfacial interactions and unique interfacial properties, thereby opening an avenue for the development of related heterojunction materials and devices.

Interfaces in functional materials and devices have attracted extensive and continuing interest since they often act as the birthplace of fascinating physical/chemical phenomena and properties[1–5]. As well known, two-dimensional electron gas, interfacial superconductivity, interfacial luminescence, and interfacial magnetism were discovered at interfaces between two cystals[6–10]. Such intriguing interfacial phenomena and properties are usually attributed to the strong physical and chemical interactions at the interfaces, thus most of them emerges at coherent and semi-coherent interfaces. Incoherent interfaces rarely exhibit unique interfacial properties due to the very weak interfacial interactions, which makes them of very limited use and of less interest.

Generally speaking, the interfacial mismatch increases from coherent to semi-coherent and to incoherent interfaces, which accordingly leads to different misfit accommodation mechanisms at interfaces[11–14]. As indicated by the schematic diagrams in Fig. S1, the coherent interface with a small mismatch has a perfect atom-by-atom matching interfacial structure since the interfacial mismatch is accommodated by the elastic deformation of two adjacent crystal lattices. The semi-coherent interface with a moderate mismatch compensates the interfacial mismatch by introducing periodic misfit dislocations[9,15,16]. For the incoherent interface, the very large lattice mismatch cannot be compensated by misfit dislocations[17]. The adjacent crystals are rigidly stacked together by maintaining their respective lattices without deformation. In contrast to coherent and semi-coherent interfaces, incoherent interfaces are conventionally considered to have very weak interfacial interactions due to worse

[1]Shenyang National Laboratory for Materials Science, Institute of Metal Research, Chinese Academy of Sciences, School of Material Science and Engineering, University of Science and Technology of China, Shenyang 110016, China. [2]Center for the Structure of Advanced Matter, School of Electronic Engineering, Guangxi University of Science and Technology, Liuzhou 545006, China. [3]Ji Hua Laboratory, Foshan 528200, China. [4]Bay Area Center for Electron Microscopy, Songshan Lake Materials Laboratory, Dongguan 523808, China. [5]Institute of Physics, Chinese Academy of Sciences, Beijing 100190, China. [6]These authors contributed equally: Xuexi Yan, Yixiao Jiang. ✉e-mail: clchen@imr.ac.cn; xlma@imr.ac.cn

lattice matching and lack of interfacial bonding states[12,18]. However, incoherent interfaces widely exist in materials and devices[19–22]. To some extent, they are more common than coherent and semi-coherent interfaces. The conventional concepts regarding incoherent interfaces make their researches and applications neglected. Although many functional films are grown epitaxially on incoherent substrates with large mismatches, few efforts have been devoted to the investigation of atomic/electronic structures of incoherent interfaces and their applications[23–26].

AlN is an outstanding wide band gap semiconductor with excellent thermal conductivity and electrical insulation, which make it attractive for applications in optoelectronics and microwave devices[27–29]. AlN is now regarded as the most promising material for ultra-deep ultraviolet light-emitting diodes and an indispensable buffer material for GaN-based devices[30,31]. For the growth of high-quality AlN films or single crystals, $Al_2O_3$ (0001) substrates are the best choice at present[32]. Thus, deep understanding of the AlN/$Al_2O_3$ (0001) interface is of significant value to improve the growth process of AlN-based materials and promote the practical applications of AlN devices. Moreover, since both Al-N and Al-O bonds have very strong electronic interactions, the AlN/$Al_2O_3$ (0001) interface with a large mismatch (i.e., ~12%) is an idea model system to explore incoherent interfaces with strong interfacial interactions, which will contribute greatly to interface science if successful.

Here, single-crystal AlN films are grown epitaxially on $Al_2O_3$ (0001) substrates by pulsed laser deposition (PLD). Atomic and electronic structures of the incoherent AlN/$Al_2O_3$ (0001) interface are systematically investigated by advanced transmission electron microscopy (TEM), and first-principles calculations. It is revealed that the AlN/$Al_2O_3$ incoherent interface exhibits unexpected strong interfacial interactions, which leads to the formation of misfit dislocation networks, interface reconstruction, and the reduction of band gap at the interface. Cathodoluminescence (CL) spectroscopy indicates that this incoherent interface can generate a strong interfacial ultraviolet light emission.

## Results and discussion

High-resolution X-ray diffraction (HRXRD) pattern, Raman spectrum, and atomic force microscopy (AFM) image in Supplementary Figures S2 and S3 suggest that as-prepared AlN thin film is composed of wurtzite AlN with a flat surface. X-ray photoelectron spectroscopy (XPS) and atomic energy-dispersive X-ray spectroscopy (EDS) in Supplementary Fig. S4 indicates the as-prepared AlN film has a good chemical stoichiometry. Supplementary Fig. S5 shows the HRXRD reciprocal space map (RSM) and Phi-scan patterns, and the rocking curve of the AlN thin film. It is revealed that the AlN film has been fully relaxed along the out-of-plane direction, but has a small in-plane tensile strain. The AlN film grows epitaxially on the $Al_2O_3$ substrate with 30° in-plane rotation to minimize the lattice mismatch at the interface via domain matching epitaxy (DME)[33]. The FWHM is measured to be 0.015°, suggesting that the AlN film has an extremely high crystallinity.

Figure 1 shows the microstructure of the AlN/$Al_2O_3$ (0001) interface from two orthogonal directions. As can be seen from the cross-sectional TEM in Fig. 1a, the AlN film grows epitaxially on the $Al_2O_3$ (0001) substrate. The AlN/$Al_2O_3$ interface is sharp and abrupt without precipitates or secondary phases. However, bright and dark contrast regions (i.e., labeled I and II, respectively) appear alternately at the interface, indicating the existence of stress concentration. The corresponding selected area electron diffraction (SAED) pattern in Fig. 1b suggests that the orientation relationship between film and substrate is (0001)AlN//(0001)$Al_2O_3$ and [11$\bar{2}$0]AlN//[10$\bar{1}$0]$Al_2O_3$[33]. Due to the large lattice mismatch, the diffraction spots of AlN and $Al_2O_3$ are separated by a long distance, implying the incoherent nature of the interface[32,34,35]. Figures 1c and 1d are the typical plan-view TEM image and corresponding SAED pattern showing the microstructure of the AlN/$Al_2O_3$ interface. Misfit dislocation networks are formed at the AlN/$Al_2O_3$ incoherent interface, which are rarely observed in other conventional incoherent interfaces. The interfacial misfit dislocations are along three equivalent $Al_2O_3$ [10$\bar{1}$0], [1$\bar{1}$00], and [0$\bar{1}$10] directions. Based on weak beam diffraction analyses shown in Supplementary Fig. S6, the Burger vectors of the interfacial misfit dislocations are

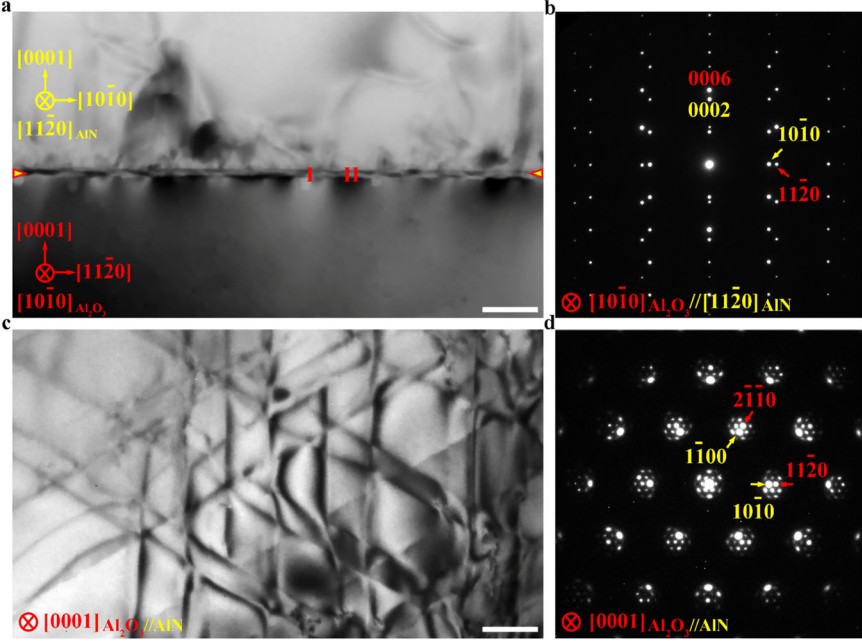

**Fig. 1 | Microstructure of the AlN/$Al_2O_3$ heterointerface. a,b** Cross-sectional TEM image (**a**) and corresponding SAED pattern (**b**) along the [11$\bar{2}$0]AlN//[10$\bar{1}$0]$Al_2O_3$ zone axis. The AlN film is epitaxially grown on the $Al_2O_3$ substrate. Bright and dark contrast regions (i.e., labeled I and II, respectively) appear alternately at the interface, indicating the existence of stress concentration. The interface is indicated by yellow arrows. **c, d** Plan-view TEM image (**c**) and corresponding SAED pattern (**d**) along the [0001] zone axis. Misfit dislocation networks are formed at the interface. The interfacial misfit dislocations are along three equivalent $Al_2O_3$ [10$\bar{1}$0], [1$\bar{1}$00], and [0$\bar{1}$10] directions. Scale bar, 200 nm.

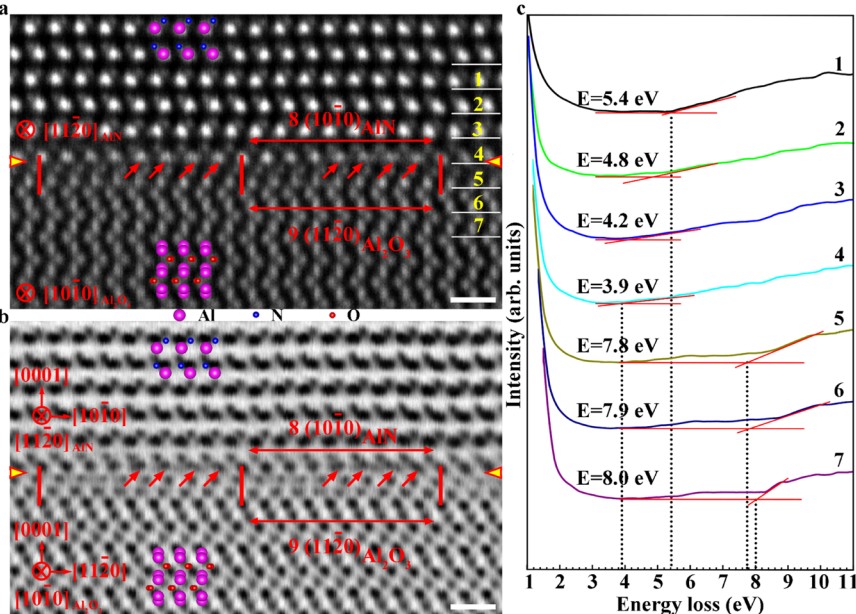

**Fig. 2 | Atomic and electronic structures of the type I interfacial structure of the AlN/Al₂O₃ incoherent interface. a, b** HAADF and ABF STEM images along the [11$\bar{2}$0]AlN//[10$\bar{1}$0]Al₂O₃ zone axis. Across the interface, the Al-terminated atomic layer of AlN is directly bonded with the O-terminated atomic layer of Al₂O₃. The AlN and Al₂O₃ lattices are rigidly stacked together by eight AlN (10$\bar{1}$0) atomic planes matching with nine Al₂O₃ (11$\bar{2}$0) atomic planes. Interfacial reconstruction and splitting of Al atomic columns can be seen clearly, as denoted by red arrows. **c** Atomic-layer-resolved valence EELS spectra across the interface. Numbers of 1-7 indicate the regions for EELS measurements. The band gap at the interface is significantly smaller than those of the bulk materials on both sides. E represents the band-gap energy. Scale bar, 5 Å.

determined to be 1/3<10$\bar{1}$0>, suggesting that the misfit dislocation networks are composed of 60° partial dislocations.

To reveal the atomic and electronic structures of the AlN/Al₂O₃ incoherent interface, atom-resolved scanning transmission electron microscopy (STEM) imaging and atomic-layer-resolved valence electron energy-loss spectroscopy (EELS) analysis were carried out. Typically, two types of interfacial structures, corresponding to the type I and type II regions in Fig. 1a, were obtained. Figure 2a is a typical high-angle annual dark-field (HAADF) STEM image showing the atomic structure of the type I interfacial structure. As can be seen, the interface is atomically abrupt and flat. Across the interface, the Al-terminated atomic layer of AlN is directly bonded with the O-terminated atomic layer of Al₂O₃. There is no evident elastic deformation of lattices in this region. The AlN and Al₂O₃ lattices are rigidly stacked together by eight AlN (10$\bar{1}$0) atomic planes matching with nine Al₂O₃ (11$\bar{2}$0) atomic planes. These features are consistent with those of conventional incoherent interfaces. However, a closer inspection reveals that interfacial reconstruction occurs at the transient Al atomic plane due to the N-Al-O bonds, which leads to the splitting of one Al atomic columns into two adjacent columns, as denoted by the red arrows. The interfacial reconstruction and splitting of Al atomic columns can be seen more clearly in the annular bright-field (ABF) STEM image in Fig. 2b, in which we can easily identify the extra Al atomic columns, as marked by the red arrows. The interfacial reconstruction has led to the shift of the band gap at the interface. Figure 2c shows the atomic-layer-resolved valence EELS spectra across the interface. As one can see, the band gap at the interface is measured to be 3.9 eV, which is significantly smaller than the intrinsic band gaps of bulk AlN and Al₂O₃ (i.e., 5.4 eV and 8.0 eV, respectively).

The type II regions at the interface have a more complex interfacial atomic structure compared to the type I regions. The atomic and electronic structures of the type II interfacial structure are shown in Fig. 3. As shown in the HAADF and ABF STEM images along the [11$\bar{2}$0] AlN//[10$\bar{1}$0]Al₂O₃ zone axis (Figs. 3a and 3b), an interfacial stacking fault is formed in the Al₂O₃ side at the interface. For better

understanding, the stacking fault is indicated by dashed yellow line in Supplementary Fig. S7. However, the formation of interfacial stacking fault does not change the lattice mismatch between AlN and Al₂O₃. Similar as the type I structure, the type II interfacial structure also has no evident elastic deformation of lattices. It is clear that eight AlN (10$\bar{1}$0) atomic planes match with nine Al₂O₃ (11$\bar{2}$0) atomic planes across the interface. The interface with the type II interfacial structure still remains the incoherent nature. Atomic-layer-resolved valence EELS analyses were also performed to determine the band gaps across the interface. As shown in Fig. 3c, the band gap at the interface is also 3.9 eV, which is greatly smaller than those of the bulk materials. The transition between the type I and type II interfacial structures is shown in the HAADF image in Supplementary Fig. S8. It is clear that the type I and type II interfacial structures are connected together by a misfit dislocation with a projected Burgers vector of 1/6[11$\bar{2}$0]. To determine the chemical distribution across the AlN/Al₂O₃ interface, atomic-layer-resolved EELS spectra of the O and N K-edges across the type I and type II interfacial structures are shown in Supplementary Figs. S9 and S10. As one can see, the AlN film is terminated with Al plane and the Al₂O₃ substrate is terminated with O plane at the interface. The AlN/Al₂O₃ interface has no detectable anion intermixing.

The formation of interfacial misfit dislocations and stacking faults, the interfacial reconstruction, and the reduction of band gap, all prove that there are very strong interfacial interactions at the AlN/Al₂O₃ incoherent interface. To reveal the origin for the strong interfacial interactions at the interface, first-principles calculations were carried out. On the basis of atomic-resolution HAADF and ABF images in Fig. 2 and Fig. 3, several possible atomic models of the type I and type II interfacial structures were built for first-principles calculations. After structural relaxation, the atomic models were used for STEM image simulations. The obtained atomic models of the type I and type II interfacial structures are shown in Figs. 4a and 4d, respectively, and their simulated HAADF and ABF images are shown in Supplementary Fig. S11. The simulated HAADF and ABF images are consistent well with the experimental counterparts.

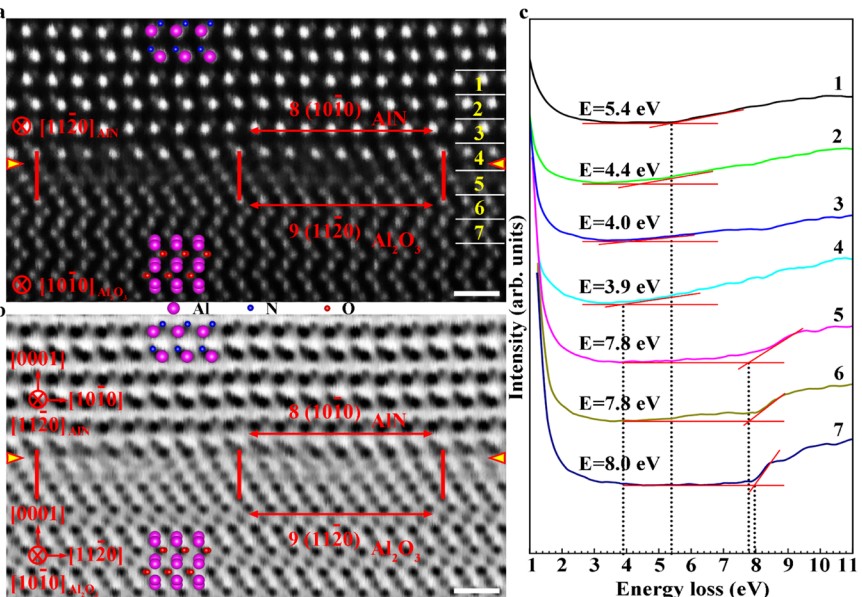

**Fig. 3 | Atomic and electronic structures of the type II interfacial structure at the interface. a, b** HAADF and ABF STEM images along the $[11\bar{2}0]AlN//[10\bar{1}0]Al_2O_3$ zone axis. An interfacial stacking fault is formed in the $Al_2O_3$ side at the interface. However, the formation of interfacial stacking fault does not change the lattice mismatch between AlN and $Al_2O_3$. Eight AlN $(10\bar{1}0)$ atomic planes match with nine $Al_2O_3$ $(11\bar{2}0)$ atomic planes across the interface. **c** Atomic-layer-resolved valence EELS spectra across the interface. Numbers of 1-7 indicate the regions for EELS measurements. The band gap at the interface is 3.9 eV, which is greatly smaller than those of the bulk materials. E represents the band-gap energy. Scale bar, 5 Å.

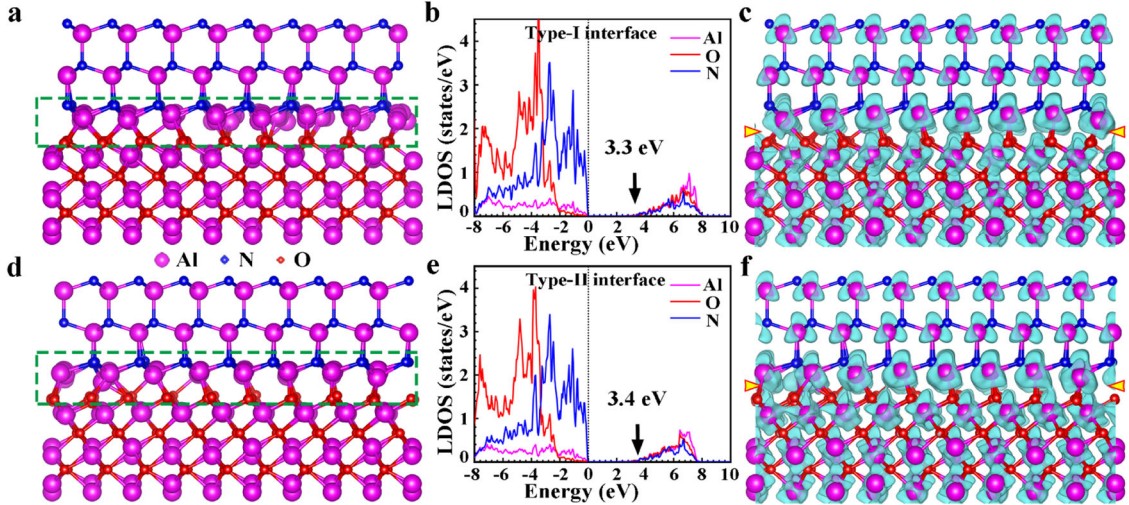

**Fig. 4 | Atomic and electronic structures of the AlN/Al₂O₃ incoherent interface. a-c** Atomic model, LDOS, and charge different densities of Al atoms of the type I interfacial structure. **d-f** Atomic model, LDOS, and charge different densities of Al atoms of the type II interfacial structure. The LDOS are plotted for the regions in the green dashed lines. Interfaces are indicated by yellow arrows. The band gaps of the type I and type II interfacial structures are calculated to be 3.3 eV and 3.4 eV, respectively. Al, N and O orbitals are overlapped at the top of valence band and the bottom of conduct band, indicating that the N-Al-O bonds are formed across the type I and type II interfacial structures of the AlN/Al₂O₃ incoherent interface. The shapes of isosurfaces of charge different densities in AlN are tetrahedral and those in Al₂O₃ are octahedral. At the type I and type II interfacial structures of the AlN/Al₂O₃ interface, the shapes of isosurfaces are between the tetrahedral and octahedral configurations, indicating a competition between the Al-N and Al-O bonds.

As shown in Fig. 4a, the interfacial Al atoms locate in the AlN₃O tetrahedra and AlN₃O₃ octahedra. Most of the interfacial tetrahedra and octahedra are distorted, which result in the relative displacement of Al atoms. Some Al atoms in severely distorted tetrahedra prefer to shift to the adjacent octahedra, which are corresponding to the extra Al atomic columns denoted by the red arrows in Fig. 2. Figures 4b and 4e show the corresponding LDOS of the type I and type II interfacial structures. The band gaps of the type I and type II interfacial structures are calculated to be 3.3 eV and 3.4 eV, respectively, which are significantly smaller than the calculated band gaps of the bulk AlN (4.5 eV) and Al₂O₃ (6.5 eV) as shown in Supplementary Fig. S12. The reduction of the calculated band gaps at the AlN/Al₂O₃ incoherent interfaces is in accord with the EELS results. The plots of LDOS show that the Al, N and O orbitals are overlapped at the top of valence band and the bottom of conduct band, indicating that the N-Al-O bonds are formed across the AlN/Al₂O₃ incoherent interfaces, thereby resulting in the strong interfacial interactions. To evaluate the strength of the interfacial interactions, the adhesion energies of the AlN/Al₂O₃

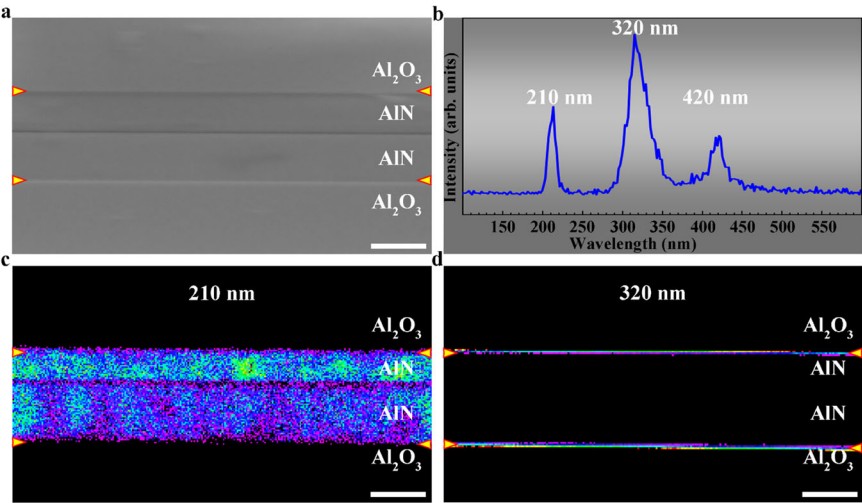

**Fig. 5 | Cathodoluminescence measurements at the AlN/Al$_2$O$_3$ interface. a** SEM secondary electron image, **b** Corresponding CL spectrum. **c**, **d** CL mappings using the 210 nm and 320 nm lasers, respectively. Interfaces are indicated by yellow arrows. The photoexcitation at 210 nm occurs throughout the whole AlN films, while the strong photoexcitation at 320 nm comes only from the AlN/Al$_2$O$_3$ interfaces, indicating the existence of a strong interfacial ultraviolet light emission at the incoherent interfaces. Scale bar, 2 μm.

interface and corresponding Al$_2$O$_3$ (0001) atomic layer in the bulk Al$_2$O$_3$ and AlN (0001) atomic layer in the bulk AlN are calculated. The adhesion energies of the type I and type II interfacial structures of the Al$_2$O$_3$/AlN interface are calculated to be −8.20 J/m$^2$ (Type I) and −8.32 J/m$^2$ (Type II), while those in the bulk Al$_2$O$_3$ and the bulk AlN are −12.63 J/m$^2$ and −7.39 J/m$^2$, respectively. The adhesion energies of type I and type II interfacial structures are even higher than that of the bulk AlN. This fact proves that the incoherent AlN/Al$_2$O$_3$ (0001) interface exhibits very strong interfacial interactions.

The charge different densities are calculated to explore the relationship between the reduction of the band gaps and the N-Al-O bonds across the type I and type II interfacial structures of the AlN/Al$_2$O$_3$ interface. The charge different densities of Al atoms are shown in Figs. 4c and 4f. The shapes of isosurfaces of charge different densities in AlN are tetrahedral and those in Al$_2$O$_3$ are octahedral, reflecting that the tetrahedral configuration of Al-N bonds in AlN and the octahedral configuration of Al-O bonds in Al$_2$O$_3$. At the type I and type II interfacial structures of the interface, the shapes of isosurfaces are between the tetrahedral and octahedral configurations due to the formation of distorted AlN$_3$O tetrahedra and AlN$_3$O$_3$ octahedra, indicating a competition between the Al-N and Al-O bonds. As a result, the Al-N and Al-O bonds of the type I and type II interfacial structures are elongated compared to those in the bulk AlN and Al$_2$O$_3$. The averaged lengths of Al-N and Al-O bonds at the interface are 1.91 Å and 1.99 Å, while those in the bulk are 1.89 Å and 1.93 Å. The Al-N and Al-O bonds at the interface are elongated by 1.0% and 3.1%, respectively. Thus, the elongated Al-N and Al-O bonds at the AlN/Al$_2$O$_3$ interface induce the reduction of the band gaps, where the Al-O bonds should make the main contribution.

To explore the potential applications of the AlN/Al$_2$O$_3$ incoherent interface, photoexcitation properties of the interface were investigated by CL spectroscopy. Figure 5a shows the cross-sectional SEM image of the AlN films bonded together in a face-to-face setup. The AlN/Al$_2$O$_3$ interfaces are indicated by yellow arrows. Figure 5b shows the corresponding CL spectrum, which exhibits three distinct emission peaks at 210 nm, 320 nm, and 420 nm, respectively. The 210 nm peak comes from the intrinsic excitation of the AlN film[36]. The 420 nm peak is an octave peak of the intrinsic excitation due to the scattering of the PMT detector grating. To determine the origin of the emission peaks, CL mappings using the 210 nm, 320 nm, and 420 nm lasers were obtained. As shown in Fig. 5c, the photoexcitation at 210 nm occurs throughout the whole AlN films, which confirms that the 210 nm peak

is the intrinsic emission of the AlN film. The CL mapping in Supplementary Fig. S13 confirms that the 420 nm peak is an octave peak of the AlN intrinsic excitation. More interestingly, as shown in Fig. 5d, the strong photoexcitation at 320 nm comes only from the AlN/Al$_2$O$_3$ interfaces, indicating that a strong interfacial ultraviolet light emission is obtained at the incoherent interfaces. The intensity of interfacial ultraviolet light emission at 320 nm is much stronger than the light emission of AlN films at 210 nm (Fig. 5b). Thus, the strong interfacial ultraviolet light emission at the AlN/Al$_2$O$_3$ interface is promising for applications in optoelectronics, photocatalysis, and biotechnology[37].

In summary, incoherent interfaces with large mismatches are traditionally believed to have very weak interfacial interactions and lack of unique interfacial properties. In contrast to these conventional views, this study has demonstrated that the incoherent AlN/Al$_2$O$_3$ (0001) interface with a large mismatch exhibits unexpected strong interfacial interactions, which lead to the formation of interfacial misfit dislocations, interfacial reconstruction, and the significant reduction of band gap at the interface. The band gap reduction is mainly due to the competition between the elongated Al-N and Al-O bonds at the interface. As a result, the incoherent AlN/Al$_2$O$_3$ interface can be excited to generate a very strong interfacial ultraviolet light emission. The discovery of strong interfacial interactions and interfacial ultraviolet light emission at the incoherent AlN/Al$_2$O$_3$ interface will not only attract scientific interest to reveal novel interfacial structures and physical properties at incoherent interfaces, but also encourage efforts to explore the potential applications of heterojunctions and devices based on incoherent interfaces with large mismatches.

## Methods

### Materials, microscopic observations, and cathodoluminescence measurements

The epitaxial AlN thin films were grown on (0001) Al$_2$O$_3$ substrates via PLD at 800 °C[38]. Prior to deposition, the Al$_2$O$_3$ substrates were annealed at 1500 °C for 3 hours in air to obtain a flat and O-terminated surface, and then transferred to the deposition chamber. The laser energy and frequency ablated on the stoichiometric AlN target under a high vacuum of 1×10$^{-4}$ Pa were 3 J/cm$^2$ and 5 Hz, respectively. After deposition for 10 hours, the AlN films were slowly cooled down to room temperature at a rate of 300 °C/hour under a N$_2$ partial pressure of 10$^2$ Pa. Thin-foil samples for TEM and STEM observations were prepared by conventional methods, including cutting, mechanical

grinding, dimpling, and Ar ion-milling. The accelerating voltage and incident angle of Ar ion beam were applied from 4.5 keV to 0.3 keV and 8° to 4°, respectively. Bright-field TEM images and SAED patterns were recorded using Tacnai F20 (FEI) and JEM 2100 (JEOL Co., Ltd) TEMs. HAADF and ABF images were taken under 300 kV by aberration-corrected STEM (Titan Cubed Themis G2300, FEI) equipped with a probe corrector (CEOS, Gmbh). For valence EELS measurements, a 200 kV STEM (ARM 200F, JEOL) equipped with a Gatan Enfina system was applied and the energy resolution of EELS spectra was 0.4 eV. We simulated the HAADF and ABF images using Dr.probe package developed by Dr. Barthel J., which is based on the multislice method[39]. The parameters for simulations were set up according to STEM experiments, where the probe convergence angle, collection semiangle, and Cs were ~ 25 mrad, 68-200 mrad, and 0.02 mm, respectively. For the photoexcitation performance measurement, cross-sectional SEM samples were prepared and mechanically polished to obtain a flat surface. Cathodoluminescence (CL) spectra were measured using a CL spectrometer (HORIBA MP-32S, Japan) equipped in a field-emission SEM (HITACHI SU-70, Japan). The SEM images were acquired under the accelerating voltage of 5 kV. The collection range of CL spectra was 100 – 600 nm by a photomultiplier detector.

## First-principles calculations

First-principles calculations based on density functional theory (DFT) were performed using Vienna ab initio simulation package (VASP) code[40–43]. Perdew-Burke-Ernzerhof (PBE) form of generalized gradient approximation (GGA) was adopted in the exchange-correlation functional[44–46]. For the plane wave expansion, projector augmented wave (PAW) pseudopotential method was used with an energy cut-off of 500 eV[47,48]. Monkhorst–Pack k-point meshes in the Brillouin zones were $6 \times 6 \times 4$ for the AlN bulk, $4 \times 4 \times 2$ for the $Al_2O_3$ bulk and $1 \times 1 \times 1$ for supercell models of $AlN/Al_2O_3$ interfaces[49]. The interfaces were relaxed until the Hellmann-Feynman forces in the system were less than 10 meV/Å. The heterojunction structures of the $AlN/Al_2O_3$ interfaces were built in 24.98 Å × 24.98 Å × 30 Å supercell models with 1052 atoms and 10 Å thick vacuum layers. The in-plane lattice parameter of the supercell models was equal to $3\sqrt{3}a_1$ and fixed during the structure optimization, where $a_1$ was the lattice parameter of $Al_2O_3$ bulk ($a_1 = 4.81$ Å and $c_1 = 13.12$ Å). The lattice parameters of AlN bulk were $a_2 = 3.13$ Å and $c_2 = 5.02$ Å. To make $8a_2$ mach $3\sqrt{3}a_1$ in the supercell models, $a_2$ was compressed by −0.2%. The adhesion energy ($E_{ad}$) of the $AlN/Al_2O_3$ interface is calculated by the equation

$$E_{ad} = [E_{tot}(AlN/Al_2O_3) - E_{tot}(Al_2O_3) - E_{tot}(AlN)]/S, \quad (1)$$

where $E_{tot}(Al_2O_3/AlN)$ denotes the total energy of $AlN/Al_2O_3$ heterojunction model, $E_{tot}(Al_2O_3)$ and $E_{tot}(AlN)$ denote the total energies of the individual $Al_2O_3$ and AlN slabs composing the $AlN/Al_2O_3$ heterojunction, and $S$ denotes the area of the $AlN/Al_2O_3$ interface.

## Data availability

The presented data were available front the corresponding author upon request.

## Code availability

The computer code that supports the findings of this study is available from the corresponding authors upon request.

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

## Acknowledgements
This study was supported by the National Natural Science Foundation of China (Nos. 52125101 (C.C.), 51971224 (C.C.), 52001309 (Y.J.)), Jihua Laboratory (Project No. X210141TL210 (H.Y.)), the Basic and Applied Basic Research Major Programme of Guangdong Province, China (Grant No. 2021B0301030003 (H.Y.)), and the Key Research Program of Frontier Sciences, CAS (QYZDY-SSW-JSC027 (C.C.)). The calculations were carried out on TianHe-1(A) at the National Supercomputer Center in Tianjin. The authors thank Dr. Jingping Cui, Yan Liang, and Lixin Yang of Institute of Metal Research for support in experiments.

## Author contributions
X.Y. and Y.J. performed the experiments, analyzed data, performed the DFT calculations, and wrote the paper. Q.J., T.Y., and W.W. help to perform TEM experiments. A.T., C.G., and X.L. help to analyzed data. C.C., H.Y., and X.M. directed the entire study. All authors read and commented on the paper.

## Competing interests
The authors declare no competing interests.
