## [Peer Review File · Nature Communications]

REVIEWER COMMENTS

Reviewer #1 (Remarks to the Author):

In this article, the authors report the unexpected strong interfacial interaction at the incoherent AlN/AO interface and further explore their applications in the ultraviolet detection. The authors had made a substantial effort in the microstructural characterizations and cathodoluminescence spectroscopy measurements. The results are of high-quality and logically discussed. Therefore, I would like to recommend it to be published in Nat. Commun. after minor revisions. The following points must be addressed in the revised manuscript.

1: In Figure 5, the 210 nm and 320 nm light-emitting peaks in CL spectra are proved to be emitted from the whole film and the heterointerface, respectively. Is there any experimental evidence to prove the 420 nm peak from the film, instead of the heterointerface?

2: More detailed information about the AlN thin film is required, such as the strain state (i.e., RSM) and composition (i.e., EDS, XPS, etc).

3 : To evaluate the strong interfacial interaction of the AlN/Al₂O₃ heterointerface, the adhesion energy of this interface must be calculated.

Reviewer #2 (Remarks to the Author):

This is a very interesting and well-researched report on the origin of strong interfacial luminescence at the incoherent AlN/Al₂O₃ interface. It is important to see that a true atomic-level explanation for the effect is presented, and opens up the possibilities for UV laser devices. It is especially nice to see this development since dislocations in wide band gap materials have long been known to be sources of light emission, but this was also regarded mostly as a curiosity. For example, dislocations in diamond (for example S.J. Pennycook, L.M. Brown, and A.J. Craven, "Observation of cathodoluminescence at single dislocations by STEM," *Philosophical Magazine A* 41: 4 (1980) 589–600). Also the following special issue of *Scanning* has many examples, for example S.J. Pennycook, "Investigating the optical properties of dislocations by scanning transmission electron microscopy," *Scanning* 30: 4 (2008) 287–298. Similarly, non-radiative transitions at interfacial dislocations in the nitride semiconductor system is a key

limitation of optical device efficiency. It would be useful to cite some of these applications to show the actual practical importance of this phenomenon.

Some comments on the terminology and English:

Throughout stack faults should be stacking faults.

Overall the paper is well written, but many phrases are slightly grammatically wrong, eg "It is revealed that the AlN/Al₂O₃ incoherent interface exhibit unexpected". Exhibit should be plural. It would improve readability of these issues could be addressed.

Reviewer #3 (Remarks to the Author):

Yan et al. report on the microscopic structural, electronic, and optical properties at the AlN/Al₂O₃ (0001) interface. The primary interests of this study span in a wide range from the atomic-scale tailoring of unconventional interface states to contributions to the development of nitrides-based optoelectronics. In addition, this interface indicates special importance as it is a seminal system for developing the field of nitride heteroepitaxy and inorganic solid-state lighting technology. Therefore, it is worth to reinvestigate this system with using advanced characterization techniques, and here the authors use STEM imaging with both ADF/ABF modes combined with spatial-resolved EELS and cathode luminescence spectroscopies. The experimental and simulated results are clear to highlight the properties of an oxide/nitride interface. However, there are a number of general issues that are not sufficiently addressed, including lack of information about chemical composition and bonding properties (especially for anion), a viewpoint on a framework of domain matching epitaxy (or higher-order epitaxy), macroscopic crystallinity, effects of anion intermixing and polarization of wurtzite lattices, and so on. The novelty of this study does not deserve a publication of nature communications. It can be published in more technical journals if the following issues are addressed.

1. Title starts from the term of string interfacial interaction, but the definition of interfacial interaction is unclear. If interfacial interaction means the strength of overall chemical bonding, how can it be verified? What is the boundary between weak and strong interactions, and how is their relative strength measured?

2. There is no information of chemical bonding and anion composition. How do EELS spectra of oxygen and nitrogen K-edges change across the interface and/or depending on local bandgaps. Anion intermixing often take place at heterointerface. Without verifying anion composition, none of the interface abruptness, interfacial atomic model for calculation, and oxynitrides polyhedra can be discussed.

3. Notion of domain matching epitaxy (DME) was established more than two decades ago (see reviews, J. Appl. Phys. 93, 278 (2003) for example), and 8/9 matching is known for the AlN/Al₂O₃ interface. This DME provides not only incoherent interfaces, but also coherent domains and equally spaced misfit dislocations (MDs) due to residual mismatch (RM) (here RM is <0.5% and MD spacing is >100 nm). MD networks seen in a plan view image (Fig. 1c) well supports that notion of DME is highly relevant to their study.

4. It is very important to show the degree of macroscopic crystallinity (such as two theta peak profiles, crystallographic tilting and twisting, surface morphology taken by AFM, etc), because all the results are microscopic and limited to local area. None of them is useful for judging sample quality.

5. PLD grown AlN layers on sapphire may be terminated with Al face (+c polarity) (see J. Cryst Growth 237-239 (2002) 1153, for example). This situation disagrees with the Al termination of AlN layer at the interface (again the lack of anion composition matters). There is no evidence to assume -c polarity (N-terminated surface), and even mixed polarity is realistic as different types of interfacial atomic structures exist in their sample.

6. EELS and CL spectra capture not only continuous states of electronic bands, but also local states of defects (here likely anions deficiencies and/or intermixing). This issue is perhaps related to question #2. Moreover, the authors need to pay attention on common behaviours of oxynitrides's bandgap: their bandgaps are almost always narrower than those of end members with large bowing parameters. This is a reason why people attempt to apply oxynitrides for solar-harvesting photoelectrodes

First of all, we would like to thank the reviewers for reviewing our manuscript very carefully and putting forth many constructive comments. We have revised our manuscript thoroughly based on these comments. Changes, including minor ones not related to the reviewers' comments, are indicated in red fonts in the revised main text and Supporting Information. Our responses to specific comments follow below:

Reply to referee 1 (R1):

We appreciate the general recognition by the referee that “In this article, the authors report the unexpected strong interfacial interaction at the incoherent AlN/A₂O₃ interface and further explore their applications in the ultraviolet detection. The authors had made a substantial effort in the microstructural characterizations and cathodoluminescence spectroscopy measurements. The results are of high-quality and logically discussed. Therefore, I would like to recommend it to be published in Nat. Commun. after minor revisions”.

In the meanwhile, the referee also raises some specific questions and comments which are summarized into three major aspects. We fully understand the referee's concerns, and here we address all the questions and discuss all the comments one-by-one in the following.

Question and comment (R1.1): In Figure 5, the 210 nm and 320 nm light-emitting peaks in CL spectra are proved to be emitted from the whole film and the heterointerface, respectively. Is there any experimental evidence to prove the 420 nm peak from the film, instead of the heterointerface?

Reply to Question and comment (R1.1):

Thank you very much for your valuable suggestion. According to your suggestions, we perform 420 nm CL-mapping, as shown in Supplementary Figure S13. It can be clearly seen that the 420 nm light-emitting peak comes from the AlN film, instead of the interface. The 420 nm peak is an octave peak of the 210 nm light-emitting peak generated by the scattering of the PMT detector grating.

We added Supplementary Figure S13 in the revised Supplementary Information.

Supplementary Figure S13 | Cathodoluminescence measurements at the AlN/Al₂O₃ interface. (a) SEM secondary electron image, **(b)** Corresponding CL mapping using the 420 nm laser. The photoexcitation at 420 nm originates from the AlN films. Scale bar, 2 μm.

Question and comment (R1.2): More detailed information about the AlN thin film is required, such as the strain state (i.e., RSM) and composition (i.e., EDS, XPS, etc).

Reply to Question and comment (R1.2):

Thank you for your useful suggestions. To provide more detailed information about the AlN thin film, we characterized the chemical composition of the AlN film by XPS and EDS, as shown in Supplementary Figure S4. Supplementary Figure S4a, b show the XPS spectra of Al 2p and N 1s on the AlN film, respectively. As one can see, both the Al 2p and N 1s peaks are very smooth, which indicate that the AlN thin film satisfies the stoichiometric ratio. The O 1s XPS spectrum cannot be detected on the AlN film. Supplementary Figure S4c, d show the atomic-resolution EDS mappings of the AlN film obtained along the [0001] zone axis. Al and N atoms are homogeneously distributed in the film. Quantitative analyses using the XPS and EDS techniques reveal that the AlN films have a good stoichiometric ratio.

Supplementary Figure S5a, b show the HRXRD reciprocal space map (RSM) patterns of the AlN film along the in-plane and out-of-plane directions. According to the RSM patterns, the lattice constants of the AlN film along the out-of-plane direction is consistent with those of perfect AlN crystal, while the in-plane lattice constants are slightly increased. These facts indicate that the AlN film has been fully relaxed along the out-of-plane direction, but has a small in-plane tensile strain. To show the in-plane matching relationship between the AlN film and Al₂O₃ substrate, phi-scan XRD has been performed (Supplementary Figure S5c). The AlN film grows epitaxially on the Al₂O₃ substrate with 30° in-plane rotation to minimize the lattice mismatch at the interface. Supplementary Figure S5d shows the rocking curve of the AlN (0002) peak. The FWHM is measured to be 0.015°, suggesting that the AlN film has an extremely high crystallinity.

Supplementary Figures S4 and S5 are added in the revised Supplementary Information.

Supplementary Figure S4 | (a,b) XPS spectra of Al 2p and N 1s on the AlN film. Both the Al 2p and N 1s peaks are very smooth, indicating that the AlN thin film satisfies the stoichiometric ratio. The O 1s XPS spectrum cannot be detected on the AlN film. (c,d) Atomic-resolution EDS mappings of the AlN film obtained along the [0001] zone axis. Al and N atoms are homogeneously distributed in the film. Quantitative analyses using the XPS and EDS techniques reveal that the AlN film has a good stoichiometric ratio. Scale bar, 5 Å.

Supplementary Figure S5 | (a,b) HRXRD reciprocal space map (RSM) patterns of the AlN film along the in-plane and out-of-plane directions. According to the RSM patterns, the lattice constants of the AlN film along the out-of-plane direction is consistent with those of perfect AlN crystal, while the in-plane lattice constants are slightly increased. These facts indicate that the AlN film has been fully relaxed along the out-of-plane direction, but has a small in-plane tensile strain. **(c)** Phi-scan patterns of the AlN film and Al₂O₃ substrate. The AlN film grows epitaxially on the Al₂O₃ substrate with 30° in-plane rotation to minimize the lattice mismatch at the interface. **(d)** Rocking curve of the AlN (0002) peak. The FWHM is measured to be 0.015°, suggesting that the AlN film has an extremely high crystallinity.

Question and comment (R1.3): To evaluate the strong interfacial interaction of the AlN/Al₂O₃ heterointerface, the adhesion energy of this interface must be calculated.

Reply to Question and comment (R1.3):

We thank the referee for the precious suggestion. The adhesion energy (E_{ad}) of the AlN/Al₂O₃ interface is calculated by the equation

$$E_{ad} = [E_{tot}(\text{AlN}/\text{Al}_2\text{O}_3) - E_{tot}(\text{Al}_2\text{O}_3) - E_{tot}(\text{AlN})]/S,$$

where $E_{tot}(\text{Al}_2\text{O}_3/\text{AlN})$ denotes the total energy of AlN/Al₂O₃ heterojunction model, $E_{tot}(\text{Al}_2\text{O}_3)$ and

$E_{\text{tot}}(\text{AlN})$ denote the total energies of the individual Al_2O_3 and AlN slabs composing the AlN/ Al_2O_3 heterojunction, and S denotes the area of the AlN/ Al_2O_3 interface. The adhesion energies of the corresponding Al_2O_3 (0001) atomic layer in the bulk Al_2O_3 and AlN (0001) atomic layer in the bulk AlN are also calculated to evaluate the magnitude of the adhesion energy of AlN/ Al_2O_3 interface. The adhesion energies of the type I and type II interfacial structures of the Al_2O_3 /AlN interface are calculated to be -8.20 J/m^2 (Type I) and -8.32 J/m^2 (Type II), while those in the bulk Al_2O_3 and the bulk AlN are -12.63 J/m^2 and -7.39 J/m^2 , respectively. The adhesion energies of type I and type II interfacial structures are even higher than that of the bulk AlN. This fact proves that the incoherent AlN/ Al_2O_3 (0001) interface exhibits very strong interfacial interactions.

The following two paragraphs are added in the revised manuscript:

To evaluate the strength of the interfacial interactions, the adhesion energies of the AlN/ Al_2O_3 interface and corresponding Al_2O_3 (0001) atomic layer in the bulk Al_2O_3 and AlN (0001) atomic layer in the bulk AlN are calculated. The adhesion energies of the type I and type II interfacial structures of the Al_2O_3 /AlN interface are calculated to be -8.20 J/m^2 (Type I) and -8.32 J/m^2 (Type II), while those in the bulk Al_2O_3 and the bulk AlN are -12.63 J/m^2 and -7.39 J/m^2 , respectively. The adhesion energies of type I and type II interfacial structures are even higher than that of the bulk AlN. This fact proves that the incoherent AlN/ Al_2O_3 (0001) interface exhibits very strong interfacial interactions.

The adhesion energy (E_{ad}) of the AlN/ Al_2O_3 interface is calculated by the equation

$$E_{\text{ad}} = [E_{\text{tot}}(\text{AlN}/\text{Al}_2\text{O}_3) - E_{\text{tot}}(\text{Al}_2\text{O}_3) - E_{\text{tot}}(\text{AlN})] / S, \quad (1)$$

where $E_{\text{tot}}(\text{Al}_2\text{O}_3/\text{AlN})$ denotes the total energy of AlN/ Al_2O_3 heterojunction model, $E_{\text{tot}}(\text{Al}_2\text{O}_3)$ and $E_{\text{tot}}(\text{AlN})$ denote the total energies of the individual Al_2O_3 and AlN slabs composing the AlN/ Al_2O_3 heterojunction, and S denotes the area of the AlN/ Al_2O_3 interface.

We sincerely thank the reviewer for the precious comments, suggestions and questions. Your comments and important questions really help us improve this manuscript.

Reply to referee 2 (R2):

Question and comment (R2.1): This is a very interesting and well-researched report on the origin of strong interfacial luminescence at the incoherent AlN/Al₂O₃ interface. It is important to see that a true atomic-level explanation for the effect is presented, and opens up the possibilities for UV laser devices. It is especially nice to see this development since dislocations in wide band gap materials have long been known to be sources of light emission, but this was also regarded mostly as a curiosity. For example, dislocations in diamond (for example S.J. Pennycook, L.M. Brown, and A.J. Craven, “Observation of cathodoluminescence at single dislocations by STEM,” *Philosophical Magazine A* 41: 4 (1980) 589 – 600). Also the following special issue of *Scanning* has many examples, for example S.J. Pennycook, “Investigating the optical properties of dislocations by scanning transmission electron microscopy,” *Scanning* 30: 4 (2008) 287 – 298. Similarly, non-radiative transitions at interfacial dislocations in the nitride semiconductor system is a key limitation of optical device efficiency. It would be useful to cite some of these applications to show the actual practical importance of this phenomenon.

Reply to Question and comment (R2.1):

Thank you very much for your recognition and high evaluation on this study. Thanks for your precious suggestion, the following references are cited in the revised manuscript:

25. Pennycook S. J., Brown L. M. & Craven A. J. Observation of cathodoluminescence at single dislocations by STEM. *Philos. Mag. A* **41**, 589–600 (1980).
26. Pennycook S. J. Investigating the optical properties of dislocations by scanning transmission electron microscopy. *Scanning* **30**, 287–298 (2008).

Question and comment (R2.2): Some comments on the terminology and English: Throughout stack faults should be stacking faults. Overall the paper is well written, but many phrases are slightly grammatically wrong, eg "It is revealed that the AlN/Al₂O₃ incoherent interface exhibit unexpected". Exhibit should be plural. It would improve readability if these issues could be addressed.

Reply to Question and comment (R2.2):

Sincerely thank you for this comment. We have carefully revised the manuscript to improve readability.

Reply to referee 3 (R3):

Question and comment (R3.1): Yan et al. report on the microscopic structural, electronic, and optical properties at the AlN/Al₂O₃ (0001) interface. The primary interests of this study span in a wide range from the atomic-scale tailoring of unconventional interface states to contributions to the development of nitrides-based optoelectronics. In addition, this interface indicates special importance as it is a seminal system for developing the field of nitride heteroepitaxy and inorganic solid-state lighting technology. Therefore, it is worth to reinvestigate this system with using advanced characterization techniques, and here the authors use STEM imaging with both ADF/ABF modes combined with spatial-resolved EELS and cathode luminescence spectroscopies. The experimental and simulated results are clear to highlight the properties of an oxide/nitride interface.

Reply to Question and comment (R3.1):

We thank the referee very much for these positive comments.

Question and comment (R3.2): However, there are a number of general issues that are not sufficiently addressed, including lack of information about chemical composition and bonding properties (especially for anion), a viewpoint on a framework of domain matching epitaxy (or higher-order epitaxy), macroscopic crystallinity, effects of anion intermixing and polarization of wurtzite lattices, and so on. The novelty of this study does not deserve a publication of nature communications.

Reply to Question and comment (R3.2):

Thanks for your detailed questions related to the study of the AlN/Al₂O₃ (0001) interface. We will address all questions one by one in this reply and add the valuable information in the revised manuscript.

However, we cannot agree with your comment about the novelty of this study. First, interfaces are hot and important topics and many important discoveries have recently reported in top journals. Interfaces in functional materials and devices have attracted intense and continuing interest since they often act as the birthplace of fascinating physical/chemical phenomena and properties. For example, E. R. Hoglund et al. (*Nature* **2022**, 601, 556-561) reported interface vibrational structure of oxide superlattices. R. S. Qi et al. (*Nature* **2021**, 599, 399-403) investigated the phonon dispersion at the cubic boron nitride/diamond interface. C. J. Liu et al. (*Science* **2021**, 371, 716-721) revealed two-dimensional superconductivity and anisotropic transport at KTaO₃ (111) interfaces. Second, our knowledge of incoherent interfaces are very limited. The intriguing interfacial phenomena and properties are usually attributed to the strong physical and chemical interactions at the interfaces, thus most of them emerges at coherent and semi-coherent interfaces. Though Incoherent interfaces are more common than coherent and semi-coherent ones, incoherent interfaces with large mismatches are often believed to be of very limited use and of less interest since they rarely generate unique interfacial

properties due to the very weak interfacial interactions. This work has changed this traditional understanding. The discovery of strong interfacial interactions and interfacial ultraviolet light emission at the incoherent AlN/Al₂O₃ interface will not only attract scientific interest to reveal novel interfacial structures and physical properties at incoherent interfaces, but also encourage efforts to explore the potential applications of heterojunctions and devices based on incoherent interfaces. Thus, we believe the novelty of this study is high.

Question and comment (R3.3): Title starts from the term of strong interfacial interaction, but the definition of interfacial interaction is unclear. If interfacial interaction means the strength of overall chemical bonding, how can it be verified? What is the boundary between weak and strong interactions, and how is their relative strength measured?

Reply to Question and comment (R3.3):

Thank you very much for your questions. When two crystals are jointed together, there are structural, physical, and chemical changes at the interface, which are called interface interactions. Though there are no absolute boundaries between weak and strong interfacial interactions, there are several empirical criteria for evaluating them:

(1) Lattice mismatch accommodation at interfaces. The coherent interface with a small mismatch has a perfect atom-by-atom matching interfacial structure since the interfacial mismatch is accommodated by the elastic deformation of two adjacent crystal lattices. The semi-coherent interface with a moderate mismatch compensates the interfacial mismatch by introducing periodic misfit dislocations. These facts indicate that the adjacent crystals of coherent and semi-coherent interfaces accommodate the lattice mismatch by modulating the crystal constants and local atomic structures near the interfaces. So the interfacial interactions at coherent and semi-coherent interfaces are strong. In contrast, the adjacent crystals of an incoherent interface are rigidly stacked together by maintaining their respective lattices without deformation. In other words, the crystal on one side of the incoherent interface has no effect on the lattice constants and atomic structure of the adjacent crystal on the other side. From this point of view, the interfacial interactions at the incoherent interface are weak. In this study, interface reconstruction occurs and misfit dislocations are formed at the incoherent AlN/Al₂O₃ interface. Thus, the interfacial interactions at the AlN/Al₂O₃ interface are very strong.

(2) Chemical bonding and new interfacial properties at interfaces. From the viewpoint of electronic structure, interfaces with strong interfacial interactions will form strong chemical bonding and generate new interfacial properties. In this study, the chemical bonds across the AlN/Al₂O₃ interface are very strong covalent bonds. The band gap at the interfaces are significantly reduced compared to those of the bulk AlN and Al₂O₃. Strong interfacial ultraviolet light emission occurs at this interface. Therefore, the interfacial interactions at the AlN/Al₂O₃ interface are very strong.

(3) Adhesion energy at interfaces. Adhesion energy at interfaces is an important factor for

evaluating the strength of interfacial interactions. The larger the adhesion energy, the stronger the interfacial interactions. To confirm the AlN/Al₂O₃ incoherent interface with strong interfacial interactions, we have added the theoretical calculations on the adhesion energy of these incoherent interfaces. The adhesion energy (E_{ad}) of the AlN/Al₂O₃ interface is calculated by the equation

$$E_{ad} = [E_{tot}(\text{AlN}/\text{Al}_2\text{O}_3) - E_{tot}(\text{Al}_2\text{O}_3) - E_{tot}(\text{AlN})]/S,$$

where $E_{tot}(\text{Al}_2\text{O}_3/\text{AlN})$ denotes the total energy of AlN/Al₂O₃ heterojunction model, $E_{tot}(\text{Al}_2\text{O}_3)$ and $E_{tot}(\text{AlN})$ denote the total energies of the individual Al₂O₃ and AlN slabs composing the AlN/Al₂O₃ heterojunction, and S denotes the area of the AlN/Al₂O₃ interface. The adhesion energies of the corresponding Al₂O₃ (0001) atomic layer in the bulk Al₂O₃ and AlN (0001) atomic layer in the bulk AlN are also calculated to evaluate the magnitude of the adhesion energy of AlN/Al₂O₃ interface. The adhesion energies of the type I and type II interfacial structures of the Al₂O₃/AlN interface are calculated to be -8.20 J/m² (Type I) and -8.32 J/m² (Type II), while those in the bulk Al₂O₃ and the bulk AlN are -12.63 J/m² and -7.39 J/m², respectively. It can be seen the adhesion energies of the Al₂O₃/AlN interfaces are between those in the bulk Al₂O₃ and the bulk AlN. These facts prove that the incoherent AlN/Al₂O₃ (0001) interfaces exhibit very strong interfacial interactions since their adhesion energies are even higher than that of the bulk AlN.

In summary, all these empirical criteria confirm that the incoherent AlN/Al₂O₃ (0001) interfaces exhibit very strong interfacial interactions.

Question and comment (R3.4): There is no information of chemical bonding and anion composition. How do EELS spectra of oxygen and nitrogen K-edges change across the interface and/or depending on local bandgaps. Anion intermixing often take place at heterointerface. Without verifying anion composition, none of the interface abruptness, interfacial atomic model for calculation, and oxynitrides polyhedra can be discussed.

Reply to Question and comment (R3.4):

Thank you for this comment and question. To verify anion composition, we measured the EELS spectra of oxygen and nitrogen K-edges layer by layer across the interface. These results are added as Supplementary Figures S9 and S10 in the revised version. The layer-by-layer EESL spectra suggest that both the type I and type II interfacial structures of the incoherent AlN/Al₂O₃ (0001) interface have no detectable anion intermixing. The interface is abrupt and the interfacial atomic models for calculations are reasonable.

The following figures (i.e., Supplementary Figure S9 and Figure S10) are added in the revised Supplementary information.

Supplementary Figure S9 | (a) HAADF STEM image of the type I interfacial structure of the AlN/Al₂O₃ incoherent interface. (b) Atomic-layer-resolved EELS spectra of the O and N K-edges across the interface. The AlN film is terminated with Al plane and the Al₂O₃ substrate is terminated with O plane at the AlN/Al₂O₃ interface. The type I interfacial structure has no detectable anion intermixing.

Supplementary Figure S10 | (a) HAADF STEM image of the type II interfacial structure of the AlN/Al₂O₃ incoherent interface. (b) Atomic-layer-resolved EELS spectra of the O and N K-edges across the interface. The AlN film is terminated with Al plane and the Al₂O₃ substrate is terminated with O plane at the AlN/Al₂O₃ interface. The type II interfacial structure has no detectable anion intermixing.

Question and comment (R3.5): Notion of domain matching epitaxy (DME) was established more than two decades ago (see reviews, *J. Appl. Phys.* 93, 278 (2003) for example), and 8/9 matching is known for the AlN/Al₂O₃ interface. This DME provides not only incoherent interfaces, but also coherent domains and equally spaced misfit dislocations (MDs) due to residual mismatch (RM) (here RM is <0.5% and MD spacing is >100 nm). MD networks seen in a plan view image (Fig. 1c) well supports that notion of DME is highly relevant to their study.

Reply to Question and comment (R3.5):

Thank you for this comment. Domain matching epitaxy (DME) is a successful paradigm for explaining the epitaxial growth of thin films. It can also reasonably interpret the orientation relationship in the AlN/Al₂O₃ system. Thanks for your suggestion, we cite the mentioned literature in the revised manuscript as the reference No. 33.

33. Narayan J. & Larson B. C. Domain epitaxy: A unified paradigm for thin film growth. *J. Appl. Phys.* 93, 278–285 (2003).

However, this study focuses on revealing the strong interfacial interaction and interfacial ultraviolet light emission at the incoherent AlN/Al₂O₃ interface. The DME theory cannot be applied in this research topic. Incoherent interfaces with large mismatches are traditionally believed to have very weak interfacial interactions and lack of unique interfacial properties. This work has changed this traditional understanding.

Question and comment (R3.6): It is very important to show the degree of macroscopic crystallinity (such as two theta peak profiles, crystallographic tilting and twisting, surface morphology taken by AFM, etc), because all the results are microscopic and limited to local area. None of them is useful for judging sample quality.

Reply to Question and comment (R3.6):

Thank you very much for your comments and suggestions. To show the degree of macroscopic crystallinity, Supplementary Figures S2, S3, S4, and S5 are added in the revised version.

Supplementary Figure S2 is an AFM image showing the surface morphology of the AlN thin film. The surface roughness obtained by this AFM measurement is 0.19 nm. This fact suggests that the surface of the AlN film is very flat.

Supplementary Figure S3 shows the XRD two theta profiles and the Raman spectrum of the AlN thin film. These results suggest that the epitaxial AlN thin film is composed of wurtzite AlN and has a good crystallinity.

To provide more detailed information about the AlN thin film, we characterized the chemical composition of the AlN film by XPS and EDS, as shown in Supplementary Figure S4. Supplementary Figure S4a, b show the XPS spectra of Al 2p and N 1s on the AlN film, respectively. As one can see,

both the Al 2p and N 1s peaks are very smooth, which indicate that the AlN thin film satisfies the stoichiometric ratio. The O 1s XPS spectrum cannot be detected on the AlN film. Supplementary Figure S4c, d show the atomic-resolution EDS mappings of the AlN film obtained along the [0001] zone axis. Al and N atoms are homogeneously distributed in the film. Quantitative analyses using the XPS and EDS techniques reveal that the AlN film has a good stoichiometric ratio.

Supplementary Figure S5a, b show the HRXRD reciprocal space map (RSM) patterns of the AlN film along the in-plane and out-of-plane directions. According to the RSM patterns, the lattice constants of the AlN film along the out-of-plane direction is consistent with those of perfect AlN crystal, while the in-plane lattice constants are slightly increased. These facts indicate that the AlN film has been fully relaxed along the out-of-plane direction, but has a small in-plane tensile strain. To show the in-plane matching relationship between the AlN film and Al₂O₃ substrate, phi-scan XRD has been performed (Supplementary Figure S5c). The AlN film grows epitaxially on the Al₂O₃ substrate with 30° in-plane rotation to minimize the lattice mismatch at the interface. Supplementary Figure S5d shows the rocking curve of the AlN (0002) peak. The FWHM is measured to be 0.015°, suggesting that the AlN film has an extremely high crystallinity.

Supplementary Figure S2 | Topographic AFM image obtained by the tapping mode showing the surface morphology of the AlN thin film. The surface roughness of the film is about 0.19 nm.

Supplementary Figure S3 | (a) Out-of-plane HRXRD pattern of the as-prepared AlN thin film on the Al₂O₃ (0001) substrate. (b) Raman spectrum of the as-prepared AlN thin film. As-prepared AlN thin film is composed of wurtzite AlN and has a good crystallinity.

Supplementary Figure S4 | (a,b) XPS spectra of Al 2p and N 1s on the AlN film. Both the Al 2p and N 1s peaks are very smooth, indicating that the AlN thin film satisfies the stoichiometric ratio. The O 1s XPS spectrum cannot be detected on the AlN film. (c,d) Atomic-resolution EDS mappings of the AlN film obtained along the [0001] zone axis. Al and N atoms are homogeneously distributed in the film. Quantitative analyses using the XPS and EDS techniques reveal that the AlN films have a good stoichiometric ratio. Scale bar, 5 Å.

Supplementary Figure S5 | (a,b) HRXRD reciprocal space map (RSM) patterns of the AlN film along the in-plane and out-of-plane directions. According to the RSM patterns, the lattice constants of the AlN film along the out-of-plane direction is consistent with those of perfect AlN crystal, while the in-plane lattice constants are slightly increased. These facts indicate that the AlN film has been fully relaxed along the out-of-plane direction, but has a small in-plane tensile strain. **(c)** Phi-scan patterns of the AlN film and Al₂O₃ substrate. The AlN film grows epitaxially on the Al₂O₃ substrate with 30° in-plane rotation to minimize the lattice mismatch at the interface. **(d)** Rocking curve of the AlN (0002) peak. The FWHM is measured to be 0.015°, suggesting that the AlN film has an extremely high crystallinity.

Question and comment (R3.7): PLD grown AlN layers on sapphire may be terminated with Al face (+c polarity) (see J. Cryst Growth 237-239 (2002) 1153, for example). This situation disagrees with the Al termination of AlN layer at the interface (again the lack of anion composition matters). There is no evidence to assume -c polarity (N-terminated surface), and even mixed polarity is realistic as different types of interfacial atomic structures exist in their sample.

Reply to Question and comment (R3.7):

Thank you very much for this comment. As you mentioned, J. Ohta *et al.* (J. Cryst Growth 237-239 (2002)) reported that AlN grown on the Al₂O₃ substrate was terminated with Al face (+c polarity) at

the surface and with the N termination at the interface. However, recent studies revealed that the polarity of AlN films on Al₂O₃ substrates depends on the termination of substrates. Many studies (For example, S. Mohn *et al.*, *Phys. Rev. Appl.* **5**, 054004 (2016); Y. Tokumoto *et al.*, *J. Mater. Sci.* **41**, 2553–2557 (2006)) reported that the AlN films grown on the O-terminated Al₂O₃ substrates exhibit the -c polarity. It is quite reasonable since the Al atomic layer of AlN will preferentially bonded with the terminated O atomic plane of Al₂O₃. These results are consistent with our study.

In fact, the termination of Al₂O₃ substrates can be controlled by chemical etching and heat treatment (Please refer to the reference: J. Toofan & P. R. Watson, The termination of the α -Al₂O₃ (0001) surface: a LEED crystallography determination. *Surf. Sci.* **401**, 162–172 (1998)). Both Al-terminated and O-terminated Al₂O₃ substrates can be prepared using proper procedures. In our study, we performed high-temperature annealing in air to obtain the O-terminated Al₂O₃ substrates (The details are presented in the **Method** section).

Sincerely thanks for your suggestions. To determine directly the termination of AlN and Al₂O₃ at the interface, we measured the EELS spectra of O and N K-edges layer by layer across the interface. These results are added as Supplementary Figures S9 and S10 in the revised version. The layer-by-layer EELS spectra suggest that the Al-terminated atomic layer of AlN is directly bonded with the O-terminated atomic layer of Al₂O₃ at the AlN/Al₂O₃ interface.

Supplementary Figure S9 and Figure S10 are added in the revised Supplementary information.

Supplementary Figure S9 | (a) HAADF STEM image of the type I interfacial structure of the AlN/Al₂O₃ incoherent interface. (b) Atomic-layer-resolved EELS spectra of the O and N K-edges across the interface. The AlN film is terminated with Al plane and the Al₂O₃ substrate is terminated with O plane at the AlN/Al₂O₃ interface. The type I interfacial structure has no detectable anion intermixing.

Supplementary Figure S10 | (a) HAADF STEM image of the type II interfacial structure of the AlN/Al₂O₃ incoherent interface. (b) Atomic-layer-resolved EELS spectra of the O and N K-edges across the interface. The AlN film is terminated with Al plane and the Al₂O₃ substrate is terminated with O plane at the AlN/Al₂O₃ interface. The type II interfacial structure has no detectable anion intermixing.

Question and comment (R3.8): EELS and CL spectra capture not only continuous states of electronic bands, but also local states of defects (here likely anions deficiencies and/or intermixing). This issue is perhaps related to question #2. Moreover, the authors need to pay attention on common behaviours of oxynitrides's bandgap: their bandgaps are almost always narrower than those of end members with large bowing parameters. This is a reason why people attempt to apply oxynitrides for solar-harvesting photoelectrodes.

Reply to Question and comment (R3.8):

Thank you very much for this comment. As you can see above, atomic-resolution HAADF and ABF images reveal that there are no precipitates or secondary phases at the interface. The interface is atomically abrupt. The layer-by-layer EESL spectra (Supplementary Figs. S9 and S10) suggest that the AlN/Al₂O₃ (0001) interface have no detectable anion intermixing. First-principles calculations reveal that the band gap reduction is mainly due to the competition between the elongated Al-N and Al-O bonds induced by the formation of distorted AlN₃O tetrahedra and AlN₃O₃ octahedra at the interface.

We sincerely thank all the reviewers for the precious comments, suggestions and questions. Your comments and important questions really help us improve this manuscript, and encourage us continuing an in-depth and all-round research in this subject. We do hope that the detailed response given to the points made above goes some way towards addressing the comments and the revised manuscript can fit your requirements.

REVIEWERS' COMMENTS

Reviewer #1 (Remarks to the Author):

The authors had revised the manuscript according to my comments. I would like to support the publication of this manuscript.

Reviewer #2 (Remarks to the Author):

In my opinion the authors have done a fine job in responding to all the referee comments and the paper should be published

Reviewer #3 (Remarks to the Author):

I have two additional comments on replies to my previous comments.

1. (R.3.3) If there is no absolute boundary between weak and strong interfacial interactions and an adhesion energy of the Al₂O₃/AlN interfaces takes a value between those of bulk parents, the term of strong interfacial interaction is nothing special and very confusing. It must be omitted from the title.
2. (R. 3.6) Notion of DME should be mentioned in main text since this mechanism is highly relevant to heteroepitaxial systems with large lattice mismatches as I mentioned already. I disagree with the following statement because one of the major focuses in Ref. 33 is the observation of misfit dislocation networks in various conventional incoherent interfaces: surprisingly, misfit dislocation networks are formed at the AlN/Al₂O₃ incoherent interface, which are rarely observed in other conventional incoherent interfaces.

Reply to referees' questions and comments:

We would like to thank all reviewers for reviewing our manuscript very carefully and acknowledging our work. We have further revised our manuscript based on reviewer #3's suggestions, which is indicated in red font in the revised main text and Supplementary information.

Reply to referee 1 (R1):

We appreciate the general recognition by the referee that "The authors had revised the manuscript according to my comments. I would like to support the publication of this manuscript".

Reply to referee 2 (R2):

Thank you very much for your recognition and high evaluation on this study.

Reply to referee 3 (R3):

Question and comment (R3.1): (R.3.3) If there is no absolute boundary between weak and strong interfacial interactions and an adhesion energy of the $\text{Al}_2\text{O}_3/\text{AlN}$ interfaces takes a value between those of bulk parents, the term of strong interfacial interaction is nothing special and very confusing. It must be omitted from the title.

Thank you very much for your suggestion. We have revised the title of the manuscript to "**Interfacial interaction and intense interfacial ultraviolet light emission at an incoherent interface**".

Question and comment (R3.2): Notion of DME should be mentioned in main text since this mechanism is highly relevant to heteroepitaxial systems with large lattice mismatches as I mentioned already. I disagree with the following statement because one of the major focuses in Ref. 33 is the observation of misfit dislocation networks in various conventional incoherent interfaces: surprisingly, misfit dislocation networks are formed at the $\text{AlN}/\text{Al}_2\text{O}_3$ incoherent interface, which are rarely observed in other conventional incoherent interfaces.

Thank you for your comments. We added a description of DME in the main text as following:

“The AlN film grows epitaxially on the Al₂O₃ substrate with 30° in-plane rotation to minimize the lattice mismatch at the interface via domain matching epitaxy (DME)³³”. In addition, we deleted the word ‘surprisingly’.